# Prosthetic Shoulder Joint Infection by *Cutibacterium acnes*: Does Rifampin Improve Prognosis? A Retrospective, Multicenter, Observational Study

**DOI:** 10.3390/antibiotics10050475

**Published:** 2021-04-21

**Authors:** Helem H. Vilchez, Rosa Escudero-Sanchez, Marta Fernandez-Sampedro, Oscar Murillo, Álvaro Auñón, Dolors Rodríguez-Pardo, Alfredo Jover-Sáenz, Mª Dolores del Toro, Alicia Rico, Luis Falgueras, Julia Praena-Segovia, Laura Guío, José A. Iribarren, Jaime Lora-Tamayo, Natividad Benito, Laura Morata, Antonio Ramirez, Melchor Riera

**Affiliations:** 1Infectious Diseases Unit, Internal Medicine Department, Hospital Universitari Son Espases, Fundació Institut d’Investigació Sanitària Illes Balears (IdISBa), 07120 Palma de Mallorca, Spain; melchor.riera@ssib.es; 2Infectious Diseases Department, Hospital Universitario Ramón y Cajal, 28034 Madrid, Spain; rosa.escudero0@gmail.com; 3Infectious Diseases Unit, Department of Medicine, Hospital Universitario Marqués de Valdecilla-IDIVAL, 39008 Cantabria, Spain; martafersam@yahoo.es; 4Infectious Diseases Department, Hospital Universitari de Bellvitge, 08907 Barcelona, Spain; omurillo@bellvitgehospital.cat; 5Bone and Joint Infection Unit, Department of Orthopaedic Surgery, IIS-Fundación Jiménez Díaz, 28040 Madrid, Spain; alvaro.aunon@gmail.com; 6Infectious Diseases Department, Hospital Universitari Vall d’Hebron, Universitat Autònoma de Barcelona, 08035 Barcelona, Spain; mariadrp7@gmail.com; 7Unit of Nosocomial Infection, Hospital Universitari Arnau de Vilanova, 25198 Lleida, Spain; ajover.lleida.ics@gencat.cat; 8Clinical Unit of Infectious Diseases, Microbiology and Preventive Medicine, Hospital Universitario Virgen Macarena CSIC, Instituto de Biomedicina de Sevilla (IBiS), Universidad de Sevilla, 41009 Sevilla, Spain; mdeltoro@us.es; 9Infectious Diseases Unit and Clinical Microbiology, Hospital Universitario La Paz, 28046 Madrid, Spain; alicia.rico@salud.madrid.org; 10Infectious Diseases Department, Corporació Sanitària Parc Taulí, 08208 Barcelona, Spain; lfalgueras@tauli.cat; 11Clinical Unit of Infectious Diseases, Microbiology and Preventive Medicine, University Hospital Virgen del Rocio, 41013 Sevilla, Spain; juliapraena@gmail.com; 12Infectious Diseases Department, Hospital Universitario Cruces, 48903 Vizcaya, Spain; LAURA.GUIOCARRION@osakidetza.eus; 13Infectious Diseases Department, Hospital Universitario Donostia, Instituto BioDonostia, 20014 San Sebastián, Spain; JOSEANTONIO.IRIBARRENLOYARTE@osakidetza.eus; 14Infectious Diseases Unit, Internal Medicine Department, Hospital Universitario 12 de Octubre, Instituto de Investigación Hospital 12 de Octubre “i + 12”, 28041 Madrid, Spain; sirsilverdelea@yahoo.com; 15Infectious Diseases Unit, Hospital de la Santa Creu i Sant Pau-Institut d’Investigació Biomèdica Sant Pau, Departament of Medicine, Universitat Autònoma de Barcelona, 08041 Barcelona, Spain; nbenito@santpau.cat; 16Department of Infectious Diseases, Hospital Clínic of Barcelona, IDIBAPS, University of Barcelona, 08036 Barcelona, Spain; LMORATA@clinic.cat; 17Microbiologic Department, Hospital Universitari Son Espases, 07120 Palma de Mallorca, Spain; antonio.ramirez@ssib.es

**Keywords:** *Cutibacterium acnes*, prosthetic joint infection, surgical and medical treatment

## Abstract

This retrospective, multicenter observational study aimed to describe the outcomes of surgical and medical treatment of *C. acnes*-related prosthetic joint infection (PJI) and the potential benefit of rifampin-based therapies. Patients with *C. acnes*-related PJI who were diagnosed and treated between January 2003 and December 2016 were included. We analyzed 44 patients with *C. acnes*-related PJI (median age, 67.5 years (IQR, 57.3–75.8)); 75% were men. The majority (61.4%) had late chronic infection according to the Tsukayama classification. All patients received surgical treatment, and most antibiotic regimens (43.2%) included β-lactam. Thirty-four patients (87.17%) were cured; five showed relapse. The final outcome (cure vs. relapse) showed a nonsignificant trend toward higher failure frequency among patients with previous prosthesis (OR: 6.89; 95% CI: 0.80–58.90) or prior surgery and infection (OR: 10.67; 95% IC: 1.08–105.28) in the same joint. Patients treated with clindamycin alone had a higher recurrence rate (40.0% vs. 8.8%). Rifampin treatment did not decrease recurrence in patients treated with β-lactams. Prior prosthesis, surgery, or infection in the same joint might be related to recurrence, and rifampin-based combinations do not seem to improve prognosis. Debridement and implant retention appear a safe option for surgical treatment of early PJI.

## 1. Introduction

*Cutibacterium* (formerly known as *Propionibacterium*) *acnes* is an anaerobic Gram-positive bacillus and a skin commensal organism with a predilection for pilosebaceous follicles, and it was formerly considered a contaminant. Moreover, *C. acnes* has been identified as a cause of biomaterial-related infections (BRIs) involving arthroplasty, cerebrospinal fluid (CSF) shunts, and spinal instrumentation, among others [1,2,3]. In recent years, with improved diagnosis methodology, including prolonged incubation protocols, *C. acnes* has become the microorganism most frequently related to infections involving shoulder prostheses. This infection type has become an emerging problem, but the relevant data are still limited [1,4,5].

*Cutibacterium* infections are usually characterized by a paucity of classical infections or inflammation symptoms, and they are often characterized by the absence of elevated inflammatory markers [1,6].

The role of *C. acnes* in prosthetic joint infections (PJIs) might be underestimated for the following reasons: (1) it is a common contaminant of the skin; (2) it needs a special transport medium; (3) it has delayed growth (up to 14 days); (4) the cultures need to be rechecked or discarded within 3 to 5 days of incubation. The advent of matrix-assisted laser desorption ionization time-of-flight mass spectrometry (MALDI-TOF MS) for the routine diagnosis of bacterial infections in clinical laboratories has increased the speed and ease of anaerobic bacteria identification [4,7,8].

*Cutibacterium* appears to have a greater predilection for infections involving the shoulder joint compared to other anatomical regions. The risk factors for *C. acnes-*related orthopedic infection include a history of joint surgery prior to the index surgery and male sex [9,10].

*C. acnes* is usually susceptible to a wide range of common antibiotics but there are no clinical trials or extensive observational studies that allow us to know the best antibiotic regimen or surgical procedure in these patients. The Infectious Diseases Society of America (IDSA) guidelines recommend penicillin or ceftriaxone as first-line treatment for *C. acnes*-related PJIs, with clindamycin or vancomycin as alternatives, and minocycline or doxycycline for suppressive therapy [11]. However, there have also been reports of increased antimicrobial resistance in biofilm-associated *C. acnes* isolates in vitro. In vitro and animal models of *C. acnes* biofilms suggest the efficacy of rifampin against *C. acnes*-related foreign-body infections [12,13], but adjunctive rifampin therapy is not included in the IDSA recommendations for *C. acnes-*related PJI management.

Despite its antimicrobial susceptibility, *C. acnes* is sometimes remarkably difficult to eradicate; therefore, medical management of PJIs without surgical intervention has been considered to result in poorer clinical outcomes [2].

The aim of this study was to describe the epidemiological, clinical, and biological characteristics, as well as the outcomes of surgical and medical treatment, of *C. acnes*-related PJI and the potential benefit of rifampin-based therapeutic combinations.

## 2. Results

Forty-six cases of *C. acnes*-related PJI were identified, of which two patients were excluded because both had co-infections with a microorganism other than CNS. Finally, we included 44 patients with *C. acnes*-related PJI. The median patient age was 67.5 years (IQR, 57.3–75.8); 75% of the patients were men. The number of cases included, according to year, is shown in Figure 1.

### 2.1. Patient Baseline and Clinical Characteristics

Demographic data, comorbidities, risk factors predisposing to PJI, signs and symptoms, and laboratory data at presentation are shown in Table 1. Most cases were classified as late chronic infection (type 2) or positive intraoperative culture (type 4), with 25% being acute prosthetic infections according to the Tsukayama classification. However, according to the Zimmerli classification, the most frequent type of infection was early infection (52.3%), while delayed and late infections were present in 47.7% of cases.

### 2.2. Microbiological Characteristics and Antimicrobial Susceptibility Patterns

With regard to microbiological data, diagnosis was performed preoperatively and/or intraoperatively in all patients. In 17 (38.6%) of the 44 patients, *C. acnes* was found in the joint fluid aspiration. In 42 (95.5%) of 44 patients, *C. acnes* was found in intraoperative samples. There were 15 patients with *C. acnes* isolation in both samples (joint and intraoperative).

Three or more positive cultures were obtained in 32 patients (72.7%), two cultures were obtained in seven patients (15.9%), and only one culture was obtained in five patients (11.4%), where the infection was demonstrated by histopathologic inflammation and positive sonicate fluid from the prosthetic material culture. All tested isolates were susceptible to β-lactams (penicillin), vancomycin, and rifampin (Table 2).

### 2.3. Surgical and Medical Therapy

All patients received surgical treatment: two-stage procedure (38.6%), debridement and implant retention (DAIR) (36.4%), one-stage procedure (18.2%), arthrodesis (2.3%), and resection arthroplasty (4.5%). When we compared the surgical treatment received with the type of infection according to the Tsukayama classification, there was an expected association between performing DAIR and early postoperative infection (Table 3).

The majority (43.2%) of antibiotic regimens used β-lactam (amoxicillin), while clindamycin was used in 31.8% and other antibiotics (linezolid, quinolones, doxycycline, and glycopeptides—vancomycin and teicoplanin) were used in 22.7%. Rifampin was administered concurrently with at least one of the aforementioned antibiotics in 19 patients (43%), with two cases of rifampin treatment being discontinued due to adverse reactions. When we compared the type of antibiotic treatment with the type of infection, we observed no significant differences (Table 3). The median duration of antibiotic therapy was 56 days (IQR, 44–84 days).

### 2.4. Treatment Outcomes

Among the 44 patients included, 39 were evaluable for treatment outcome. At the last follow-up, five patients were lost, 34 patients were considered cured, and five had microbiologically confirmed recurrence. Three patients died due to noninfectious causes (acute pulmonary edema, advanced renal neoplasm, and cardiorespiratory arrest); these patients were followed up for more than 12 months with favorable infection outcomes.

We compared patients with a favorable outcome to those who failed treatment (Table 4). All patients in the failure group were male, but there was no significant difference in the clinical presentation, treatment received, or type of infection. A nonsignificant trend toward a higher frequency of failure was observed among patients with previous prosthesis (odds ratio (OR): 6.89; 95% confidence interval (CI): 0.80–58.90; *p* = 0.078) and previous surgery and infection in the same joint (OR: 10.67; 95% IC: 1.08–105.28; *p* = 0.043). In addition, we observed a higher frequency of recurrence in diabetic patients (OR: 4.87; 95% IC: 0.69–34.50; *p* = 0.113) and those who were treated only with clindamycin (OR: 6.89; 95% IC: 0.80–58.90; *p* = 0.078) than those who only received amoxicillin (OR: 0.357; 95% CI: 0.04–3.55; *p* = 0.379) or rifampin-based combinations (OR: 0.844; 95% CI: 0.12–5.72; *p* = 0.862).

Regarding surgical treatment, 15/39 patients (38.5%) underwent DAIR, with 13 having favorable outcomes (Figure 2). When analyzed according to both classifications, for patients classified by the Tsukayama guidelines, 9/13 cured patients (69.23%) had type 1 infections and 4/13 (30.77%) had type 2, with one case of recurrence for type 1 and another recurrence for type 2. According to the Zimmerli classification, 12/13 of the cured patients (92.30%) had early infections and 1/13 (7.7%) had a delayed infection, with both recurrences being classified as early infections. Of the 24 patients treated with prosthesis removal, only three had recurrence (12.5%) (Table 4).

Among the 39 evaluable patients, 17 were treated with rifampin. There were no differences in the outcome of patients treated with rifampin-based combinations. There were five patients with positive intraoperative cultures, with one being treated with rifampin therapy and cured, while the others (4/5) did not receive rifampin and recurrences were observed. Of the 15 patients treated with DAIR, eight (53.3%) received rifampin-based regimens, while seven did not, and one recurrence was observed in each group, but this was not significant (Figure 2). We analyzed 11 patients who received clindamycin treatment (six associated with rifampin) and there were three instances of recurrence (all isolates were susceptible to clindamycin).

The epidemiological, clinical, and treatment data of the five patients who showed recurrence are presented in Table 5.

## 3. Discussion

In this retrospective multicenter study, we described 44 patients with shoulder PJI due to *C. acnes* over a period of 14 years. The diagnosis of this infection is difficult due to the absence of classical clinical evidence, as well as the challenges associated with culturing the microorganism. In this 14 year series, we observed an increase in the number of diagnosed cases of this infection, which is probably due to the extended incubation time that has been demonstrated in other studies for maximizing the recovery of *C. acnes* from PJI specimens [1,7,14,15,16,17].

Previous studies have argued that the shoulder has a propensity for infection with *C. acnes* because it is the anaerobic dominant bacteria from healthy skin, particularly in moist areas (axilla), where a higher *C. acnes* bacterial burden is observed in men compared to women [17,18,19]. Moreover, previous series have reported that male gender is a risk factor for the development of this infection [1,9,20]. These previous findings would explain our results in which a male predominance of PJI was observed.

The most frequent types of infection in this study, according to the Tsukayama classification, were late chronic or positive intraoperative cultures, which is similar to that reported in other studies [1,21,22]; this is due to the paucity of classical symptoms and the absence of elevated inflammatory markers that delay diagnosis. However, when we classified the infection type according to the Zimmerli classification, early infection was the most frequent.

In our study, the most frequent symptom was joint pain. This is consistent with other studies in which pain and functional limitations without either fever or constitutional symptoms were the most frequent clinical presentations [6,7,23].

Previous surgery in the same joint has been linked to an increased risk of *C. acnes*-related PJI because repeated manipulation of the joint causes changes in the anatomical structure; this increases the duration of surgery, which is a major risk factor for shoulder PJI from this microorganism [20,24,25]. We observed that previous prosthesis, infection, or surgery in the same joint might be related with recurrence, but we could not demonstrate a significant association, possibly due to small sample size.

In our study, all isolates tested were susceptible to penicillin, vancomycin, and rifampin, with approximately 2.5% being resistant to clindamycin. These observed susceptibility patterns were similar to those of other studies [1,7,22], which suggested that the broad antimicrobial susceptibility of *C. acnes* appeared to be maintained.

Previous clinical studies and case reports provide little information regarding the optimal treatment for *C. acnes*-related PJI. In our study, all patients received antibiotic and surgical treatment. As expected, we observed significant differences between surgery type (DAIR, two-stage surgery, and one-stage surgery), as well as the type of prosthetic infection. Regarding surgical treatment, prosthesis retention and the two-stage procedure were the most frequent surgical procedures performed, unlike previous articles, which suggests that prosthesis exchange should be the treatment of choice in most cases [2,5,26]. We observed that, in the cases of early infection according to the Zimmerli classification, DAIR treatment may be a safe option.

In terms of antimicrobial treatment, the outcomes with or without adjunctive rifampin therapy were similar to other studies [1,22]. This finding is striking, particularly in cases treated with debridement and implant retention, because this antibiotic has antibiofilm activity and its effectiveness for the eradication of *C. acnes* has been demonstrated both in vitro and in vivo in an animal model of foreign-body infection [12]. However, another explanation could be that the presence of a high inoculum in the biofilms forms in a foreign body (i.e., a prosthetic joint). In this state, the microorganism produces mutations that can lead to some degree of resistance, which is observed as a reduced susceptibility to rifampin; this phenomenon was reported in a study by Furustrand et al. [27], where it was demonstrated in vitro.

On the other hand, we observed a nonsignificant trend toward a higher frequency of failure among 11 patients who received clindamycin (cured 72.7% vs. recurrence 27.3%). The IDSA guidelines recommend clindamycin as an alternative treatment to β-lactams, because the majority of tested isolates were susceptible; for this reason, the use of clindamycin has been evaluated in previous studies [1,2,3]. However, future clinical trials will be needed to compare antibiotic therapy between β-lactams and clindamycin in *C. acnes*-related PJI.

The strength of this multicenter study is that only patients with a proven diagnosis of *C. acnes-*related PJI were included. Currently, most studies include all types of bone infection due to this microorganism, which makes it difficult to determine the best management and evolution of this entity.

This study did have some limitations. This was a retrospective observational study that did not have predefined therapeutic procedures, and this could have induced bias. Moreover, the follow-up time was limited to a 1 year period. However, in PJIs caused by microorganisms as paucisymptomatic as *C. acnes*, in which DAIR has been performed, a longer follow-up time might be necessary.

## 4. Methods

### 4.1. Study Design, Patients, and Settings

This multicenter, retrospective observational study was conducted at 16 hospitals belonging to the Prosthetic Joint Infection Group of the Spanish Network for Research in Infectious Diseases between January 2003 and December 2016.

Patients aged 18 years and older with shoulder PJIs that were caused by *C. acnes* and diagnosed between January 2003 and December 2016 were included, regardless of the age of the implant at the time of the initial symptoms. Polymicrobial infections with coagulase-negative staphylococci (CNS) were also included.

### 4.2. Data Collection

Cases were identified by searching the databases of previously recorded consecutive PJIs or the general archives at each participating hospital.

Medical chart abstraction was performed using a standardized case report form to retrieve demographic, clinical, and laboratory data. Demographic data included age and sex. Laboratory data included erythrocyte sedimentation rate (ESR), C-reactive protein (CRP), and white blood cell (WBC) counts. Clinical data consisted of comorbidities, immunosuppressive therapy, Charlson index, previous exposure to antibiotics (7 days), hospitalization in the previous 90 days (of at least 2 days), and receipt of hemodialysis. We also collected the following information regarding arthroplasty: date of implantation, site, primary or revision arthroplasty, previous infections in the same joint (date and microorganism), cemented versus uncemented arthroplasty, use of antibiotics in bone cement, and date of diagnosis. The time from index surgery to diagnosis was recorded as the time from the last surgical procedure performed pre diagnosis to the first positive *C. acnes* culture, with classification of the PJI, type and number of cultured samples, and their results also being recorded.

Information regarding surgical treatment, exchange of removable pieces of the prosthesis (in at least one debridement surgery), and the type and duration of antimicrobials used was also collected, as well as patient outcomes and the date of the last follow-up visit.

### 4.3. Definitions

A PJI was defined on the basis of previously detailed criteria [1,11]. The *C. acnes* etiology was confirmed if ≥2 specimens were positive for *C. acnes*, or if one culture specimen was positive for *C. acnes*, with no other organism detected on culture and concurrent evidence of joint purulence, histopathological inflammation, or a sinus tract communicating with the prosthesis. PJI was assigned according to the Tsukayama and Zimmerli classifications [28,29,30].

### 4.4. Follow-Up and Treatment Success

Antimicrobial therapeutic regimens and treatment outcomes were assessed through the last recorded clinical visit. Decisions on therapeutic regimens were based on the clinical judgment of the infectious disease and surgical specialist providers. The type, delivery method, and duration of antimicrobial therapy were recorded.

After being discharged, patients were followed-up according to the protocol of each participating center. The follow-up period was calculated from surgery due to infection: debridement, one-stage exchange, two-stage exchange, or other procedures (arthrodesis/resection arthroplasty). Among patients in remission, only those with at least 1 year of follow-up were included in the outcome analysis.

Cure was defined as the absence of signs and symptoms of infection at the conclusion of a minimum 1 year follow-up period after antibiotic therapy, which did not result in unplanned additional surgical debridement for putative persistent infection. Treatment failure was established on the basis of the following criteria: (1) persistence of symptoms and clinical signs of infection during treatment that led to a change in the surgical strategy (except for new surgical debridement during the first month after an initial debridement); (2) the recurrence of symptoms and clinical signs of infection once the surgical strategy was completed, with isolation of the same microorganism; (3) the need for suppressive antibiotic treatment against *C. acnes*; (4) infection-related death. Any case of reinfection by microorganisms other than *C. acnes* detected during the follow-up period was not considered a failure.

### 4.5. Microbiological Methods

Culture specimens were collected and processed at each participating institution, following the Spanish guidelines for the microbiological diagnosis of bone and joint infections [31,32]. Identification testing of isolates was performed in the clinical microbiology laboratory at each center using standard microbiological techniques. The susceptibilities of *C. acnes* isolates were tested against standard antimicrobial agents. Isolates were classified as susceptible according to the minimum inhibitory concentration (MIC) breakpoints set by the Clinical and Laboratory Standards Institute (CLSI) or the European Committee on Antimicrobial Susceptibility Testing (EUCAST).

### 4.6. Statistical Analysis

The descriptive analysis for defining the patient’s characteristics was done by frequencies and percentages for categorical variables and measures of central tendency and dispersion for numerical variables. The non-normally distributed continuous variables were expressed by median and interquartile range (IQR). For evaluating the differences between favorable outcome and failed treatment, the Mann–Whitney *U* test was used to compare continuous variables and the chi-squared and Fisher exact tests were used for comparing categorical variables. Moreover, univariate logistic regression was used for evaluating the recurrence risk. A *p*-value <0.05 was considered statistically significant. Statistical analyses were performed using SPSS software version 26 (IBM Inc., Armonk, NY, USA).

## 5. Conclusions

Physicians should be aware of the increase in the frequency of shoulder PJIs caused by *C. acnes* because there are few clinical symptoms and an absence of elevated inflammatory markers. On the other hand, patients with type 1 infections according to the Tsukayama classification or early infection by the Zimmerli classification could be treated with DAIR. According to our data, rifampin therapy does not seem to improve outcomes, and clindamycin seems to be associated with a worse prognosis. Randomized studies with a greater number of patients are necessary to establish the optimal antimicrobial treatment.

## Figures and Tables

**Figure 1 antibiotics-10-00475-f001:**
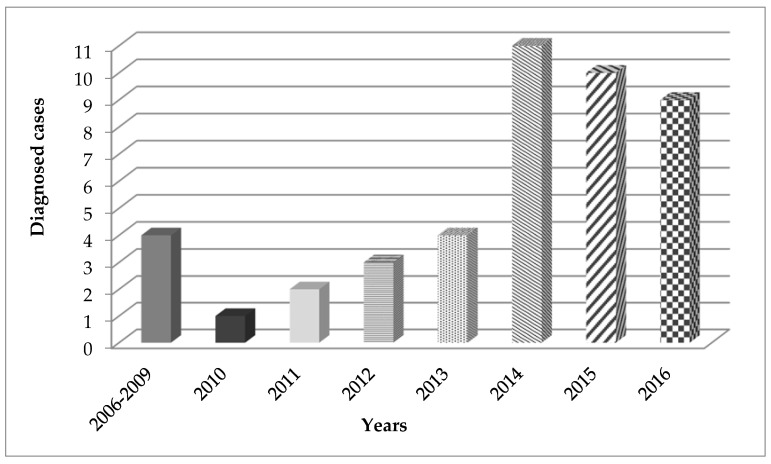
Cases frequency by year.

**Figure 2 antibiotics-10-00475-f002:**
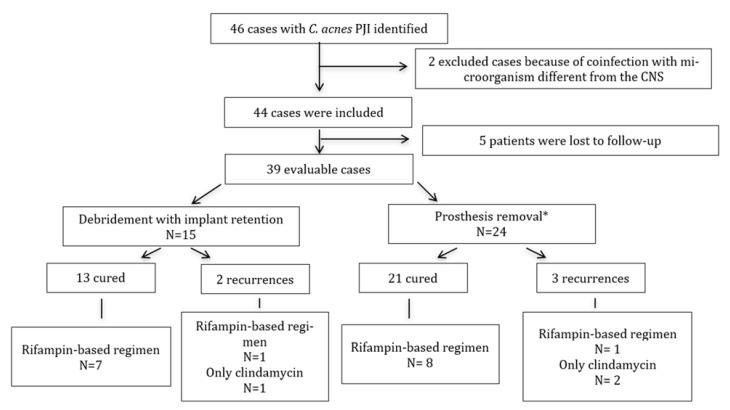
Flowchart of failure rates according to the medical and surgical approaches used. * Fifteen patients were treated with a two-stage procedure, six were treated with a one-stage procedure, one was treated with arthrodesis, and two were treated with resection arthroplasty.

**Table 1 antibiotics-10-00475-t001:** Demographic, clinical, and laboratory characteristics of shoulder PJI due to *C. acnes.*

Variable	No (%) ^a^
**Age, years ^b^**	67.5 (IQR, 57.3–75.8)
**Male**	33 (75)
**Charlson Index ^b^**	3.0 (IQR, 0.0–4.0)
**Comorbidities**	
Diabetes mellitus	13 (29.5)
Oncologic diseases	8 (18.2)
Renal insufficiency	3 (6.8)
Immunosuppressive treatment	2 (4.5)
Others	14 (31.8)
**Time to diagnosis, days ^b^**	78.0 (IQR, 10.0–431.0)
**Previous prosthesis**	5 (11.4)
**Previous surgery**	5 (11.4)
**Previous infections**	5 (11.4)
**Prosthesis infection**	
Right shoulder	24 (54.5)
Left shoulder	20 (45.5)
**Clinical characteristics**	
Fever	8 (18.2)
Joint pain	33 (75)
Swelling	23 (52.3)
Fistula	7 (15.9)
Purulent wound drainage	12 (27.3)
**Laboratory parameters ^b^**	
WBC count, cells/mm^3^	8245.0 (IRQ, 6427.5–10,367.5)
CRP, mg/dL	14.0 (IQR, 6.0–32.3)
ESR, mm/h	46.0 (IQR, 22.0–71.0)
**Type of shoulder PJI**	
**• Tsukayama classification**	
Early postoperative infection	11 (25.0)
Late chronic infection	27 (61.4)
Positive intraoperative infection	6 (13.6)
**• Zimmerli classification**	
Early infection	23 (52.3)
Delayed or low-grade infection	14 (31.8)
Late infection	7 (15.9)

Abbreviations: CRP, C-reactive protein; ESR, erythrocyte sedimentation rate; WBC, white blood cell. ^a^ Data are the number (%) of cases. ^b^ Median (IQR, interquartile ranges).

**Table 2 antibiotics-10-00475-t002:** Samples and microbiological characteristics of shoulder PJI due to *C. acnes.*

Variable	Patients No. (%) ^a^
**Samples taken for culture**	
Joint aspirate fluid	17 (38.6)
Intraoperative sample	42 (95.5)
Joint fluid + intraoperative samples	15 (34.1)
**Microorganisms isolated**	
Only *P. acnes*	35 (79.5)
Co-infection with *S. epidermidis*	9 (20.5)
**Microbial susceptibility ^b^**	
Penicillin	39 (100)
Vancomycin	27 (100)
Clindamycin	38 (97.4)
Tetracycline	13 (100)
Rifampin	23 (100)

Abbreviations: CLSI, Clinical and Laboratory Standards Institute; EUCAST, European Committee on Antimicrobial susceptibility testing. ^a^ Data are the number (%) of cases; susceptibilities determined as per CLSI/EUCAST breakpoints. ^b^ All antibiotics were not tested in all strains isolated.

**Table 3 antibiotics-10-00475-t003:** Comparison between the types of treatment with type of infection of shoulder PJI due to *C. acnes.*

Treatment	Type of Infection No. (%) ^a^			Total (*n* = 44)	*p*
	Type 4 ^b^ (*n* = 6)	Type 2 ^b^ (*n* = 27)	Type 1 ^b^ (*n* = 11)		
**Antibiotic**					
Amoxicillin	3 (50.0)	13 (48.1)	3 (27.3)	19 (43.2)	0.558
Clindamycin	3 (50.0)	8 (29.6)	3 (27.3)	14 (31.8)	0.650
Rifampin	2 (33.3)	11 (40.7)	6 (54.5)	19 (43.2)	0.677
**Surgical**					
Debridement and retention	0	6 (22.2)	10 (90.9)	16 (36.4)	0.000
2-stage procedure	1 (16.7)	15 (55.6)	1 (9.1)	17 (38.6)	0.013
1-stage procedure	4 (66.7)	4 (14.8)	0 (0)	8 (18.2)	0.006
Arthrodesis	0	1 (3.7)	0 (0)	1 (2.3)	1
Resection arthroplasty	1 (16.7)	1 (3.7)	0 (0)	2 (4.5)	0.315

^a^ Data are the number (%) of cases. ^b^ Tsukayama classification: early postoperative infection (Type 1), late chronic infection (Type 2), and positive intraoperative cultures (Type 4).

**Table 4 antibiotics-10-00475-t004:** Comparison of final outcomes.

	Outcome		
Variable	Cured (*N* = 34)	Recurrence (*N* = 5)	*p*
**Age, years** (**Median**)	68 (IQR, 57.8–76.3)	69 (IQ, 42.5–73.5)	0.378
**Gender, No.** (**%**)			
Male	24 (70.6)	5 (100)	0.302
Female	10 (29.4)	0 (0)	
**Charlson Index** (**Median**)	2.95 (IQR, 0–4.03)	2.0 (IQR, 0–4.50)	0.729
**Comorbidities, No.** (**%**)			
Diabetes	8 (23.5)	3 (60)	0.125
Renal insufficiency	2 (5.9)	1 (20)	0.345
Oncologic disease	7 (20.6)		0.563
Immunosuppressive therapy	2 (5.9)		1
**Previous prosthesis, No.** (**%**)	3 (8.8)	2 (40)	0.114
**Previous surgery, No.** (**%**)	2 (5.9)	2 (40)	0.072
**Previous infections, No.** (**%**)	2 (5.9)	2 (40)	0.072
**Prosthesis infection, No.** (**%**)			
Right shoulder	18 (53)	3 (60)	1
Left shoulder	16 (47)	2 (40)	
**Time to diagnosis, days** (**Median**)	67 (IQR, 9–199)	70 (IQR, 7–1537)	0.823
**Type of infection No.** (**%**)			
**• Tsukayama classification**			1
Early postoperative infection	10 (29.4)	1 (20.0)	
Late chronic infection	20 (58.8)	3 (60.0)	
Positive intraoperative cultures	4 (11.8)	1 (20.0)	
**• Zimmerli classification**			0.823
Early infection	19 (55.9)	3 (60)	
Delayed or low-grade infection	11 (32.4)	1 (20)	
Late infection	4 (11.8)	1 (20)	
**Surgical treatment, No.** (**%**)			
Prosthesis retention	13 (38.2)	2 (40)	1
1-stage procedure	5 (14.7)	1 (20)	1
2-stage procedure	13 (38.2)	2 (40)	1
Arthrodesis	1 (2.9)	0	1
Resection arthroplasty	2 (5.9)	0	1
**Antimicrobial treatment, No.** (**%**)			
Amoxicillin	14 (41.2)	1 (20)	0.631
Clindamycin	3 (8.8)	2 (40)	0.114
Other	2 (5.9)	0	1
Amoxicillin plus rifampin	3 (8.8)	1 (20)	0.436
Clindamycin plus rifampin	5 (14.7)	1 (20)	1
Other plus rifampin	7 (20.6)	0	0.563

**Table 5 antibiotics-10-00475-t005:** Individual clinical characteristics and treatment of the five recurrence cases.

Patient	Age, Years ^a^	Sex	Comorbid Factors	Clinical Signs and Symptoms	Delay in Diagnosis, Days	Type of Infection		Treatment		
						Tsukayama ^b^	Zimmerli	Type of Surgery	Antibiotic Regimen	Duration, Days
1	69	Male		Fever	407	Type 2	Delayed infection	1-stage procedure	Amoxicillin	57
2	74	Male	DM, CKD	Joint pain, joint swelling	70	Type 2	Early infection	DAIR	Amoxicillin plus Rifampin	138
3	52	Male	DM	Joint pain, joint swelling, fistula	2667	Type 2	Late infection	2-stage procedure	Clindamycin plus Rifampin	112
4	73	Male	DM		0	Type 4	Early infection	2-stage procedure	Clindamycin	60
5	33	Male		Joint pain, joint swelling, fistula, purulent wound drainage	14	Type 1	Early infection	DAIR	Clindamycin	175

Abbreviations: DM, diabetes mellitus; CKD, Chronic kidney disease. ^a^ Mean age (SD), 60.2 (SD 17.6) years. ^b^ Tsukayama classification: early postoperative infection (Type 1), late chronic infection (Type 2), and positive intraoperative cultures (Type 4).

## Data Availability

The data presented in this study are available on request from the corresponding author. The data are not publicly available due to ethical concerns.

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
