# Peer review of "Prosthetic Shoulder Joint Infection by Cutibacterium acnes: Does Rifampin Improve Prognosis? A Retrospective, Multicenter, Observational Study"

_antibiotics, 2021, doi:10.3390/antibiotics10050475_

Round 1
Reviewer 1 Report
I understand the importance of studies regarding infections from foreign bodies such as prosthetic joints. This study is well designed in a multicenter observational fashion.
However, the methods about the diagnosis of prosthetic joint infections (PJI), especially the definition of PJI in this study, is not clearly described. In the guideline from IDSA (Clinical Infectious Diseases 2013;56(1):e1–25), the definition of PJI is described as follows;
Definition of PJI
12. The presence of a sinus tract that communicates with
the prosthesis is definitive evidence of PJI (B-III).
13. The presence of acute inflammation as seen on histopathologic examination of periprosthetic tissue at the time of
surgical debridement or prosthesis removal as defined by
the attending pathologist is highly suggestive evidence of PJI
(B-II).
14. The presence of purulence without another known etiology surrounding the prosthesis is definitive evidence of PJI
(B-III).
15. Two or more intraoperative cultures or combination of
preoperative aspiration and intraoperative cultures that yield
the same organism (indistinguishable based on common laboratory tests including genus and species identification or
common antibiogram) may be considered definitive evidence
of PJI. Growth of a virulent microorganism (eg, S. aureus) in
a single specimen of a tissue biopsy or synovial fluid may
also represent PJI. One of multiple tissue cultures or a single
aspiration culture that yields an organism that is a common
contaminant (eg, coagulase-negative staphylococci, Propionibacterium acnes) should not necessarily be considered evidence of definite PJI and should be evaluated in the context of
other available evidence (B-III).
The authors should state clearly how much patients in their cohort shall satisfy each of the definition mentioned above.
In addition, patients with only one culture of C. acnes should be treated cautiously, and the diagnosis of PJI should be made only when other supportive evidence of PJI could be detected in the same patient. The authors should make it clear how much of patients with only one culture could satisfy the definition of PJI.
When these concerns were properly addressed this manuscript should be considered for publication.
Reviewer 2 Report
Dear Authors,
The article describes the results of a retrospective, multicenter observational study regarding prosthetic infection of the shoulder joint by the microorganism Cutibacterium acnes and the impact that incorporation of rifampicin has on treatment. Although the focus on the antimicrobial activity of rifampin is an important part of all the work, the title does not represent it, this reviewer considers that the focus on rifampin should be incorporated into the title.
The following changes to the document are suggested:
- It is more correct to describe beta-lactam or β-lactam, than “b-lactam”. Lines 52, 57, 130, 142, 285.
- Figure 1. Shows characters as squares. I do not know if it is my program or the writing was misconfigured.
- In microbiological susceptibility, it is not necessary to place the total number of patients, if the percentage is expressed in parentheses. It says 39/39 (100), it should say 39 (100), the same for the rest of antibiotics.
- In Table 3. In the Antibiotics list it is described as “rifampicin therapy”. Since the reason for this difference is not disclosed, it is suggested that simply rifampicin be placed.
- Figure 2. It is deconfigured with the line numbers.
- Table 5. Unconfigured at the end. Line 222.
This reviewer suggests that more information be provided to support the use of rifampicin in the treatment, from the point of view, of antimicrobial resistance.
Round 2
Reviewer 1 Report
The concerns raised by the reviewer were properly addressed. This article would be informative for clinicians treating PJIs.